# Leading wellness in healthcare: A qualitative study of leadership practices for wellness in hospital settings

**Julaine Allan**[1]☉*, **Katarzyna Olcon**[2]☉, **Ruth Everingham**[2,3], **Mim Fox**[2], **Padmini Pai**[3], **Maria Mackay**[4], **Lynne Keevers**[2]

**1** Rural Health Research Institute, Charles Sturt University, Orange, New South Wales, Australia, **2** School of Health and Society, University of Wollongong, Wollongong, New South Wales, Australia, **3** Illawarra Shoalhaven Local Health District, Warrawong, New South Wales, Australia, **4** Graduate School of Medicine, University of Wollongong, Wollongong, New South Wales, Australia

☉ These authors contributed equally to this work.
* juallan@csu.edu.au

**Data Availability Statement:** The underlying data (interview and focus group transcripts) from the study cannot be shared because of requirements of the approving Human Research Ethics Committee.

## Abstract

Ways of dealing with workplace stress and enhancing healthcare workers wellness are sought globally. The aim of this study was to explore healthcare leaders' practice in relation to the implementation of a workplace wellness program called SEED in the context of multiple crises (bushfires and COVID-19) affecting a local health district in New South Wales, Australia. Practice theory informed interviews (n = 23), focus groups (n = 2) and co-analysis reflexive discussions (n = 2) that were conducted with thirteen leaders and twenty healthcare workers. A pragmatic approach to program implementation for healthcare workers' wellness explored the process and actions that resulted from leadership practice in an inductive thematic analysis. Preliminary themes were presented in the co-analysis sessions to ensure the lived experiences of the SEED program were reflected and co-interpretation of the data was included in the analysis. Three key themes were identified. 1) Leading change—implementing a wellness program required leaders to try something new and be determined to make change happen. 2) Permission for wellness—implicit and explicit permission from leaders to engage in wellness activities during worktime was required. 3) Role-modelling wellness—leaders viewed SEED as a way to demonstrate leadership in supporting and caring for healthcare workers. SEED provided a platform for leaders who participated to demonstrate their leadership practices in supporting wellness activities. Leadership practices are critical to the implementation of healthcare wellness programs. The implementation of SEED at a time of unprecedented crisis gave leaders and healthcare workers opportunities to experience something new including leadership that was courageous, responsive and authentic. The study highlighted the need for workplace wellness programs to intentionally include leaders rather than only expect them to implement them. The practices documented in this study provide guidance to others developing, implementing and researching workplace wellness programs.

The participant information sheets guaranteed participants confidentiality and that no identifying information would be shared. The qualitative data was collected from a small group of participants in a specific identifiable location. Transcripts are unable to be completely anonymized/deidentified to remove information about the interviewees or the people they refer to in their discussions. Data requests may be sent to the University of Wollongong & ISLHD Health and Medical Human Research Ethics Committee (no. 2021/ETH00110) by contacting the university Human Research Ethics Team on T: +61 2 4221 3773 or E: uow-humanethics@uow.edu.au.

**Funding:** JA, KO, PP, MF, MM and LK all received funding for this study from the Australian National Health and Medical Research Council Medical Research Future Fund - COVID-19 mental health research. Grant number APP2005659. https://www.health.gov.au/summary-of-mrff-grant-recipients The funders had no role in study design, data collection and analysis, decision to publish, or preparation of the manuscript.

## Introduction

Healthcare staff work in stressful environments. They primarily deal with people who are ill or suffering and have the responsibility of ameliorating the symptoms or disease in some way. These challenging work circumstances and high workloads can interrupt healthcare workers personal lives and work capacity with physical and mental health consequences, for example burnout [1–5]. Ways of dealing with workplace stress and enhancing healthcare workers wellness are sought globally.

Workforce health in large organisations is typically the responsibility of human resources departments and addressed through health promotion campaigns that aim to change employees lifestyle risk factors outside of the workplace [6]. Most programs hope to reduce health-related costs and improve productivity[7,8]. Wellness programs conducted during work time are less common and few have been evaluated [9]. However, they primarily focus on an individual's diet, exercise and stress levels [e.g.6,8,9]. In this study workplace wellness programs are conceptualised eudemonically as activities conducted at work, designed for groups or teams, aiming to enhance belonging and purpose, and develop trusting and supportive relationships with colleagues [10].

The COVID-19 pandemic has exacerbated workplace stress in health settings. Since the pandemic was declared in March 2020 by the World Health Organization [11], it has had a profound impact on the healthcare system internationally. The pandemic has highlighted the existing stressful nature of the healthcare work environment. Many studies of the pandemic's impact call for workplace interventions to address burnout and improve wellness to sustain and retain healthcare staff [e.g. 12–14]. However, implementing any new program poses significant challenges including developing plans, policies, and structures to support implementation [15], and fostering support and positive communication around the program [16]. When the program is not directly the organisation's core business, as in the case of healthcare workers wellness, additional barriers arise such as justifying the need for and value of the innovation [17]. To ensure successful implementation and sustainability of wellness initiatives, leadership commitment is critical [18,19]. Accordingly, the purpose of this study is to analyse leadership practices during the implementation of a workplace wellness initiative in one local health district in the aftermath of the 2019–2020 Australian bushfires and COVID-19. Following an overview of healthcare leadership literature, the analysis of qualitative data from the perspectives of both leaders and healthcare workers is presented.

### Leadership in healthcare

Leadership has been defined as "a process whereby an individual influences a group of individuals to achieve a common goal" [20 p.43]. Leadership is commonly understood in terms of a hierarchy with one or more people at the top the only ones in leadership roles [20]. However, in complex and layered organisations, there are multiple teams and hierarchies with leadership roles typically defined by position names such as Manager and Team Leader. The role of leader acknowledged by a title can be different to the practice of leadership. Leadership practice "utilises the ability to understand the wider environment to identify a direction, foster a culture, motivate people and illicit their commitment" [21 p.164]. Leadership theorists have elaborated several leadership styles including transactional, transformational, and servant leadership which encourage change in different ways—from using incentives and self-interest, through inspiration, charisma and individual attention, to placing the good of followers over the leader's own self-interest [21]. Other leadership styles such as authentic and quiet leadership recognise a person's unique experiences as shaping their leadership practice and influence even if not within a formal leader position [22,23]. Regardless of ideal leadership practice, healthcare

roles are organised hierarchically with leadership responsibilities assigned to individuals in certain positions.

Healthcare leaders work in an environment characterised by the importance of quality care, managing risk, and optimising patient outcomes while minimising costs and operational inefficiencies [24]. Strict policies and processes attempt to ensure consistency across sites and meet standards such as clinical and health and safety guidelines among others. Operational complexities need to be managed while at the same time resiliently leading change that is intended to transform the system [25–27]. Therefore, capacity and capability for leadership at every level of the system is the imperative [28]. When the challenge is the implementation of a new program, leaders have a key role to play.

Nevertheless, the sole focus on the skills, qualities and behaviours of leaders and the heirarchial systems of authority within which policies and decisions are made risks ignoring the experiences and behaviours of those being led and overlooking the practice of leading [29]. Leadership occurs within an organisational culture and the most successful cultures are characterised by kindness, trust and respect that supports employee participation, creativity and productivity[30,31]. However, constant change is a consistent feature of modern workplaces resulting in staff with initiative fatigue, experiences of previous programs that have been abandoned or concerns about increased workload and lack of resources for new strategies, interventions, or processes [28,32].

Practice theory offers a way to explore ways leaders and followers contribute to implementation of new initiatives. A practice approach looks for the "doings", "sayings", "relatings", that together make up a practice [29]. Thus a practice is created by the relations between people as they connect in a setting. The practice theory literature conceptualises leadership as occurring as a practice rather than residing in the traits of individuals [2,33]. Leadership-as-practice is less about what one person thinks or does and more about what healthcare staff may accomplish together [34]. Accordingly, it is concerned with how leadership emerges, unfolds, shapes and is shaped by healthcare sites at particular times and contexts [29,33]. Thus, for this study practice theory suggests investigating leadership as situated, dynamic, context specific and experience-based.

Guided by practice theory, the aim of this study was to explore healthcare leaders' practice in relation to the implementation of a workplace wellness program called SEED in the context of multiple crises affecting a local health district. To understand leading and doing rather than leadership as possessed, the study incorporates the perspectives of both leaders and healthcare workers and focuses on horizontal and informal leadership relationships [33].

## Methods

This study is part of a larger research project that explored the wellness practices initiated at a healthcare organisation in response to bushfires and COVID-19. A qualitative approach guided by practice theory shaped the semi-structured interview guide, focus groups and co-analysis reflexive discussions. No specific questions were asked about leadership during the data collection, but it was frequently discussed by participants when talking about how SEED was implemented. Accordingly, chronicling and analysing one of the eight key practices identified in earlier analysis [19]–responsive and compassionate *leading*–was determined by the research team to be worth analysing in more depth.

This study was approved by the Ethics Committee of The University of Wollongong & ISLHD Health and Medical Human Research Ethics Committee (no. 2021/ETH00110) and the Aboriginal Health and Medical Research Council Human Research Ethics Committee (no. 1779/21). No ethical issues were identified in the study. The study was designed by all authors

except RE who was responsible for data collection, administration and contributed to analysis. JA, KO, LK, PP, and MM all have PhDs and have extensive experience in qualitative research in healthcare. LK has extensive experience in applying practice theory to research and guided the conceptual framework of the study. All authors are female with backgrounds in social work, nursing and occupational therapy.

## Study setting

The Illawarra Shoalhaven Local Health District (ISLHD) is a publicly funded healthcare provider covering 250 kilometres of the New South Wales south coast and 400,000 residents [35]. The SEED wellness program was established at a small rural hospital within ISLHD in response to bushfires in 2019–20 to alleviate psychological impact on healthcare staff. The need for an organisational response was recognised by the ISLHD CEO who had experience of working through a natural disaster [36]. Developed from a post-traumatic growth framework and using strengths-based methods, SEED took a bottom-up approach to identifying staff needs and facilitating staff-led wellbeing activities focussed on team support and collective caring for each other [19]. Wellness activities were workplace based and resourced, and voluntary to facilitate ease of access and participation. Staff co-designed the wellness program. Consultation with staff at all levels in a work team took place from the very beginning to ensure the wellness program incorporated activities they wanted to do. This included staff groups getting together, sharing their stories of stress and distress and supporting each other to design ways to address their needs. Practical examples include ensuring staff took breaks, providing quiet spaces, providing food, having yoga sessions. Critically, each site decided themselves what they wanted and how it would be done (see [37] for a detailed program description). As the COVID-19 pandemic impacted healthcare staff across the district, the SEED program was subsequently adopted in five other hospitals and administration sites during 2020 and 2021, when this study took place.

## Data collection and sample characteristics

Purposive sampling was used to recruit participants who were ISLHD employees, aged over 18 years and who had participated in SEED activities. Email contact lists provided by the SEED Program Lead were used to circulate the invitation to participate. Forty-four staff volunteered to participate. However, due to COVID-19 restrictions and workplace demands, the final number of individual participants was 33 (Table 1). Thirteen participants were in leadership roles including organisation-wide roles and site and team leaders from the five SEED implementation sites. Leaders had worked in healthcare for between 15 and 37 years. The remaining

**Table 1. Study participant demographics.**

| Gender | Mean age in years (range) | Work role (n) | Mean number of years in current role (range) |
|---|---|---|---|
| Female 29<br>Male 4 | 49.9 (32–65) | Nurse* 9<br>Nurse Educator 4<br>Senior Nurse Educator (L)^ 1<br>Manager/Executive (L) 10<br>Health and Security 2<br>Allied Health 2<br>Administration 3<br>Project manager 1<br>Senior Project Manager (L) 1 | 7.5 (1–37) |

*Nurse–includes all patient facing roles (Registered Nurse, Enrolled Nurse, Clinical Nurse Specialist). ^L denotes formal leadership role.

twenty participants were healthcare employees in nursing, allied health, security, and administration positions in the SEED sites.

Interviews (n = 23), focus groups (n = 2) and co-analysis sessions (n = 2) were conducted between May 2021 and March 2022. Data collection was conducted by site. Focus groups and co-analysis sessions included both leaders and healthcare workers who had participated in SEED. Because SEED is inclusive of all staff in a team who participate equally, data collection was not organised by role. Semi-structured interview and focus group guides developed by the research team were used to explore participant's experiences of the SEED program and perceptions of its impact. Participants were asked open-ended questions such as "How did SEED begin for you?", "What does SEED look like in your workplace?" and "What is SEED about?"

The focus groups ranged from 85 to 120 minutes in length (average of 105 minutes conducted by MF and KO). Interviews ranged from 28 to 110 minutes in length (average of 55 minutes Conducted by JA, KO, RE and MF). The researchers conducting the data collection were not known to the participants prior to the study. Participants were provided with written information about the study and given the opportunity to ask questions prior to being asked for written consent. Invitations to one hour on-line co-analysis sessions were sent to all participants by email. Two times were provided to maximise participation opportunities. There were ten participants in co-analysis session 1 and nine in session 2. All data collection sessions were recorded and professionally transcribed.

## Data analysis

A pragmatic view of the way leadership practice was enacted was taken [38]. Pragmatism in research focusses on inquiring how knowledge is turned into action by asking about what was done in implementation [4]. In data analysis, a pragmatic approach to program implementation for healthcare workers' wellness results in exploration of the process and actions that result from leadership practice rather than elaboration of leadership theory. That is, what did the leaders do and how did they relate to others [33]. Inductive thematic analysis [39] was conducted individually by the research team with five members reading twenty-four transcripts between them.

The aim of the co-analysis sessions was to to ensure participants' lived experience of the SEED program was reflected in the analysis and they had the opportunity of co-interpretation of the data [40,41]. Preliminary themes were presented to participants (n = 19) via a powerpoint presentation that summarised the study processes and the broad practices identified by the research team. Participants in the co-analysis sessions were asked questions to prompt discussion such as 'what practices stand out as effective in enhancing well-being at work?, and What do you think are the most significant practices, themes and findings that embody the SEED program? When you think about your experience of SEED, what is mssing from the data presented today? The importance of leadership as a core practice of implementing SEED emerged from these analysis processes.

All the leadership statements were compiled into NVivo and read repeatedly by two members of the research team (RE and JA). Statements from leaders and healthcare workers were interrogated for similarities and differences. Questions asked of the data included, "How was the leadership role in SEED implementation described?" "How was the role of leadership described by a) SEED participants in positions of leadership and b) by other participants who were led by them?" Leadership themes were discussed and confirmed by four members of the research team (RE, JA, KO, and MF).

## Results

Three key themes were identified from the data– 1) Leading change, 2) Permission for wellness and 3) Role-modelling wellness. Each theme had several sub-themes as detailed in Fig 1. To

**Fig 1. Summary of the findings.** Fig 1-*Leading wellness in healthcare*: *A qualitative study of leadership partices for wellness in hospital settings.*

protect the confidentiality of participants, they will be referred to as Leader (L) or Healthcare Worker (HW), and when quoted will be identified by the number assigned to them in the data management system.

## Leading change

Implementing a wellness program was described by most participants as being different and outside the usual health response to healthcare workers' needs. This required leaders to be willing to try something new and being determined to make the change happen. Because the wellness program activities were healthcare worker-led, the leaders had to have courage to relinquish control and take risks. Leading change in this way seemed to have supported informal leadership and enhanced healthcare workers' faith and trust in healthcare leaders.

**Willingness, determination and trust.** Following the natural disasters that affected the region and the subsequent COVID-19 pandemic, leaders participating in the study reported being prepared to try anything that would alleviate the healthcare workers' stress and assist in recovery. As one leader emphasised, however, they had to be willing to lead something new and different as no one knew exactly what needed to be done: "*And I think that's important too, is that leaders of the wards or the hospitals that use SEED have to be open-minded and be prepared to try something new*" (L19).

In contrast to previous frustrations of programs commencing and staff not feeling heard, the implementation of the SEED wellness program felt different. Many participants described how they appreciated the immediacy of action that they observed in the leaders at various levels to address their needs. For example:

> . . .the most important thing was that thought was given and it was instantaneously accommodated, which brought the faith and trust and my belief in 'I'm validated and heard and that, wow, someone's listening to concerns and the solution to it is happening quite quickly' (HW16).

Leaders' determination and the concrete steps they took to make change happen was perceived as unique and crucial for the successful implementation of the wellness program. Leaders described being driven by the belief in the initiative, even though they were not sure what exactly it should look like in practice: "*If I know that there's really positive intent behind something I'll go, 'We will make it happen.' And that's something that I said to [SEED Project Leader] quite a few times. I said, 'No, we'll make it happen'*" (L21).

Having trust in leadership was described by one healthcare worker to be fundamental–"*in the workplace, trust and respect are kind of like the foundations* [of strong workplace relationships]" (HW19). Leading change in a way that was open to the unknown, determined, accountable and courageous helped healthcare workers' build trust in the leadership. However, healthcare workers also had to trust that what their leaders were asking them to invest time and energy in was worthwhile, particularly when time and energy were in short supply. For example, "*just really trusting that this work is important, this work is valuable, and if we're doing it with a good intention, just trust that it'll work out*" (HW 17).

**Courage to relinquish control and take risks.**   Given the immediacy of the need for organisational response to healthcare workers' psychological distress and the crisis situation, both the processes and the intended outcomes of the SEED program were not clearly defined in the beginning. This meant leaders having the courage to relinquish the need to be in charge which was described as unusual in the healthcare setting. This willingness to take the risks and trust the process was described by a leader in the following way:

> How can you roll a program out where you don't know what it's going to look like, and you don't know the end game and you don't know what the outcome's going to be? Well, you've just got to trust that it's going to be okay. And you've got to trust that you've got the capacity to adapt to the needs of the people at the time. So that's what we did (L01).

Moreover, although leadership initiative and support were critical to the implementation of the new program, SEED activities were ultimately co-designed and healthcare worker led. This meant leaders required additional courage to relinquish the need to be in charge. As one leader explained:

> We've got that staff-led design as well–and I think that's really important, and I had the courage to say 'I don't know what I'm putting my hand up for, but I just know that I'm connected with whatever this is and that's what we need to grow' (L21).

Healthcare workers recognised the courage it took for leaders to take risks and to relinquish control over the direction SEED might take:

> The leadership of the organisation said, 'We don't know what you need to do but go and do it and work with those that know what they need'. I don't know that I had really seen that before in the organisation. So that was very courageous (L25).

Finally, in an environment characterised by risk management, the leaders acknowledged that courage was necessary to take risks in implementing SEED, but it was easier when they were supported by their own leaders:

> It's hard to be courageous if you're stepping into the space by yourself. I think wards that have done the best, it's definitely when they've had their ops manager or general manager or you know, certainly, to have your CE [Chief Executive] standing behind you can give you much more confidence (L25).

**Supporting informal leadership.** Leaders relinquishing control over the wellness program allowed informal leaders to emerge. The SEED program facilitated informal leadership because it was healthcare worker-led giving some who were not in leadership positions the opportunity to lead in wellness. The result of this was a demonstration of informal leadership and how it could operate. In the words of one leader: *"It's not just those in traditional leadership positions. What it showed in a real meaningful sense is that leadership comes at all levels"* (L32). Some perceived the valuing of the informal leader role in SEED had organisational wide benefits. For example:

> I love the fact that SEED and the leadership associated with it can be formal and informal, but essentially if I think if you do it the right way these things can start to become a core competency for the organisation you're in (L23).

In this way, leading was co-produced through the practices of SEED. Examples were provided of how the recognition of informal leaders supported the wellness activities and healthcare workers in general:

> the informal versus formal leader, the NUM's the formal leader in the ward environment, but within that, you've got a whole host of opportunistic informal leaders. I've got an enrolled nurse that works at [Hospital] who, just by his presence, is a leader. He provides leadership through his role modelling of behaviours. [It's] recognising that the informal leaders are making a difference (L01).

## Permission for wellness

Study participants frequently used the word "permission" when talking about workplace wellness and the role of leaders in it. Healthcare workers described the impact of getting both implicit and explicit permission from leaders to engage in wellness activities during worktime. Leaders similarly emphasised the importance of giving healthcare workers permission to focus on wellness in the workspace. Nevertheless, many leaders did not seem to apply the same permission for wellness to themselves. They seemed constrained in many ways to be able to look after themselves in the workplace, a trend that, as some admitted, extended beyond their work hours.

**Giving and receiving permission for wellness.** Permission to focus on wellness during work hours from leaders such as the nurse unit managers and management above them was viewed by healthcare workers as something unique and *"out of character to our workplace"* (HW 13). Nevertheless, healthcare workers felt that the permission was genuine and made it much easier for them to take their wellness seriously and engage in wellness activities. As one health worker explained:

> It's about knowing what wellness is for you internally and that might mean that you need to have 10 minutes' walk every day as part of that, but it's acknowledging that.It's being given permission to look after that in a workplace, which is really, really good. So much lip service in the past, 'you've got to look after yourself', but what did that mean? It didn't translate in practice because you were just so busy (HW29).

The healthcare workers alluded here to the frequent workplace recommendation of self-care, which is perceived simply as a "lip service" if leaders do not make self-care a priority within the work setting. Real permission to focus on wellness during work hours changed the

way some healthcare workers felt about their work environment, for example: *"you're happier to come to work if you feel like you're supported and that you're given permission to look after yourself"* (HW 29).

Leaders, on the other hand, perceived giving healthcare workers permission for wellness as something that was needed in the aftermath of a crisis. Leaders believed they had to actively encourage and support healthcare workers to participate in wellness activities:

> It was giving people permission, and I was one of those, as a leader, that went, 'We can do this. It's okay to do this. You can do this because this is what we need. This is collectively what we need, and you have my support to do it' (L21).

Giving healthcare workers permission to participate in wellness activities was described as a leadership behaviour rather than a response to a request to do something. For example, "*she empowers them with the permission to do this and demonstrates what leadership's going to look like*" (L01).

Leader support for workplace wellness, however, was not always forthcoming. Participants described a few instances where some leaders did not appear to see the need for the wellness initiatives and subsequently, directly or indirectly, withheld their permission for healthcare workers to engage in wellness activities. As one participant described: *"we've had one area that we thought that really needs something and the leader could not see the need for anything. Where the leader doesn't believe in it, it's not going to work"* (L33). Because SEED was a new local initiative, and voluntary, there was no requirement for wards or teams to participate in it. There were no policies or directives guiding the implementation. Therefore, if it did not appeal to leaders, they did not have to facilitate their team's participation. This situation is explored further in the valuing and sustaining wellness activities sub theme.

**Not giving self the permission for wellness.** Even though many of the leaders supported wellness activities and saw their value for healthcare workers, they frequently did not prioritise their own wellness. They described the stress and demands of leading healthcare as the primary barrier to participation in workplace wellness initiatives. As one leader explained:

> Sometimes I resoundingly fail at that one [participating in SEED] because of the pace we have to work in. If I looked across the system now, I'm not sure we're looking after ourselves well. We're going at many fronts. And it's night and day, and it's seven days a week. And that's what our jobs are (L33).

Many leaders described their jobs as extremely demanding leaving no time to focus on their own wellness. The job's stressors and demands were not only a daily routine during the work hours but often extended to their life outside of work. Some of the leaders in executive roles described their jobs as taking over their entire week and not leaving any time for activities such as cooking or exercising. As one leader concluded referring to wellness: "*those people that you're responsible for, they get that benefit but sometimes we don't want to allow ourselves to be the recipients at the same time*" (L21).

Another barrier to leaders' wellness was that there seemed to be no one to look after the leaders. Leaders were perceived as having the responsibility for healthcare workers' wellness, but their needs were often overlooked: *"There's no one really looking after the leaders, that I'm aware of. There's a lot of gratitude and a lot of 'thank you for the work you do', but nothing formal by way of a wellness program"* (L30). Leaders were concerned that if they did not monitor their own wellness, there was no one who would do it for them. Consequently, they believed that many leaders across the healthcare system needed support.

## Role-modelling wellness

The leaders who participated in the study had a strong belief in the importance of wellness and that SEED was a means to promote it amongst their healthcare workers. They also viewed SEED as a way to demonstrate leadership in supporting and caring for healthcare workers. Role-modelling was an important leadership strategy for promoting wellness. Multiple examples were provided about role-modelling wellness by leaders and valuing and sustaining wellness activities. Wellness-oriented leadership was in turn viewed as effective and successful leadership by many.

**Demonstrating wellness.**   Leaders were described to role-model wellness in two ways. First, they demonstrated SEED practices such as caring and kindness towards healthcare workers and each other and vulnerability. As one leader explained:

> If you've got a leader that says, 'I'm going to role-model wellbeing, I'm going to role-model kindness, and the way I'm going to do that is this, this and this', and then demonstrate that, it's much easier to permeate [wellness] down from that perspective (L01).

Interestingly, healthcare staff discussed how important demonstrating vulnerability was in engendering trust. When leaders allowed themselves to be seen as vulnerable within the workspace, the healthcare workers saw them as human beings who can be trusted. As one worker explained:

> this is an important part of leadership, is allowing yourself to be vulnerable, because that opens you up to people trusting you. I think that one of the fundamentals of being a good leader is being in a position where people feel safe to trust you. Where that doesn't happen you become the boss rather than the leader, and I think that's what SEED did (L23)

Second, leaders often simply "showed up" and participated in wellness activities organised by SEED such as "Wellness Wednesdays" and "Wellness Warriors". The leaders' physical presence in wellness activities, even if just for the first ten minutes, was meaningful for healthcare workers and symbolised leaders' endorsement of taking time off for wellness during worktime. In the words of one healthcare worker: *"because they [leaders] take the time out to do the yoga and to do the activities, the rest of the staff feel like they can take the time too and they don't have to feel guilty about it"* (HW 26).

This role-modelling of wellness by leaders was recognised and valued by healthcare workers and leaders alike. Several participants equated this type of leadership with what it meant to be a "good leader". One leader elaborated:

> What was really exciting for me it was seeing the qualities of leadership role-modelled and demonstrated, but [also] enabled by other leaders across the organisation. To me that just sings, it just celebrates what it is to be a good leader and to be a good role model (L25).

Some leaders believed that by embracing and striving to role-model wellness they have developed a better vision for themselves as leaders and grown in the process. This is how one leader described this transformation: *"For me SEED became a fantastic way of identifying within the organisation about what I felt strongly about, so I could really send a message about the type of leadership I was trying to grow and build" (*L23). Others gave examples of the support leaders could provide for each other by taking up SEED initiatives and practicing wellness. The caring and kindness was not just for leaders to demonstrate to their teams but also to other leaders–*"leaders supporting leaders, able to pick up the phone and debrief"* (L31).

**Valuing and sustaining wellness activities.** The role-modelling of wellness practices by different leaders in the organisation was viewed by healthcare workers as important for sustainability of the program. For example, *"sustainability comes from the leadership and that demonstration of being involved in this space"* (HW27). This view was supported by examples of problems when SEED was not supported by leaders noting that participating in staff wellness was not supported in healthcare in general;

> They don't invest in it themselves. I, unfortunately, think that there are some thoughts around it [wellness] as being a bit, you know, airy-fairy, you know, you're in health, you need to toughen up. . ..that mentality. I think the leadership has to support it and role-model it. You need to role-model it (L30).

One healthcare worker gave an example of when the importance of valuing wellness and facilitating activities via a leadership role was not supported: *"There was somebody else in his role that didn't value SEED. . .. It happened when he was on leave and somebody else stepped into that position, activities were stopped"* (HW17). In contrast, when wellness was valued by leaders it was described as having a broad impact. For example: *"if you've got your manager or the Director of a hospital that is fully supportive, they will be always thinking of things, how they can look after the wellness of their staff"* (HW29).

Moreover, participants believed that an important way to sustaining wellness activities in the healthcare setting is by adequate training of new leaders to ensure they perceive workplace wellness as an essential part of their job:

> We need to make sure that when we bring people on as new leaders, that they understand that wellbeing is as important as balancing the budget, as work health safety, as is clinical governance. We need to ensure that we bring that to the fore for them so they can role model from the beginning (L01).

In addition, participants believed that leaders need to role-model the emphasis on workplace wellness amongst themselves. This, again, was viewed as especially important for emerging leaders:

> I sit here thinking about an emerging leader program for nurses and midwives. . .it's so important for our young and emerging leaders to see it [SEED] role modelled, and then to understand how to implement, how to practice, how to work in their space successfully (L25).

Leaders' role was thus perceived as essential to not only to implementing a wellness initiative in healthcare but also to sustaining it.

## Discussion

This study explored healthcare leadership in relation to the implementation of a workplace wellness program called SEED. The study analysed the practices of leading from two perspectives–those leading the implementation of SEED and those being led—identifying consistency across both groups in the three key themes–leading change, permission for wellness and role-modelling wellness. SEED was developed and implemented at a time of unprecedented crisis in healthcare and the local community. Because it was new and untested, it provided a platform for leaders who participated to demonstrate their leadership practices in supporting wellness activities. Both leaders and healthcare workers described the leadership practices in

similar ways with healthcare workers providing important insights into how the practices were experienced and the impacts they had. In particular the leadership practices supported the eudemonic ideal of workplace wellness in that study participants described enhanced belonging and purpose, and the development of trusting and supportive relationships with colleagues [10].

Successful organisational cultures inherently include trust and respect to support employee engagement and productivity [30,31]. SEED facilitated leader's capacity to demonstrate trust and respect that had not previously been experienced by HCW. The organisation's executive leaders responded quickly to crisis through supporting SEED and facilitating staff participation as a solution to stress and burnout rather than continuing to investigate and "describe the problem" as is typical in healthcare [26 p.157]. Many things were not typical about SEED. Frequent reference was made to both SEED and the leadership practices being different to the expected workplace culture, even "out of character" [34]. The immediacy of the actions taken by leaders both initially and throughout the implementation of the SEED program were consistent with authentic and transformational leadership practices that are necessary for followers, in this case healthcare workers, to implement innovations [18,32]. Leadership in SEED implementation emerged and was shaped by the context of bushfires and COVID-19 on healthcare workers. These crises provided opportunities to try something new. While informal leaders emerged, it was the leadership practice of those in traditional leader positions who were able to give permission to practice wellness.

Typically program implementation in healthcare requires planning, policies, and structures [15,27]. However, being developed out of crisis and staff-led meant there were no structures or policies to guide the SEED implementation within the organisation, and this had several consequences. The co-development of SEED showed leadership as a collective endeavour where all those who were part of the interactions played some part in how leadership emerged and evolved [34]. Leadership participation was critical to SEED. Leaders who did not value wellness or see it as a way of addressing healthcare workers' stress did not have to participate, and this ultimately meant their teams did not participate. Leaders who did participate were early adopters of something not built into the system and had to establish their own ways of leading the program and facilitating staff participation, for example, using role-modelling which was consistently reported. The lack of structure and guidance in a highly regulated environment such as healthcare gave leaders the opportunity to demonstrate both quiet and authentic leadership by taking risks, demonstrating courage and relinquishing control [21,23]. The cumulative disasters provided the right eco-system for leaders to step back and practice quiet leadership, enabling staff to have the confidence and permission to participate [23]. It appears that implementing wellness allowed leadership to shine at all levels in the organisation.

However, leaders cannot do wellness on their own if there are no supportive organisational structures [33]. The theme of leaders not giving themselves permission for wellness shows this tension. Leadership practice includes fostering a commitment to a course of action. Leaders in this study demonstrated their own[16,21] commitment through role-modelling wellness participation that was clearly recognised by healthcare workers who were given both explicit and implicit permission to participate in wellness activities within the workplace.

Although leaders who participated in this study believed in and supported healthcare workers' wellness, they often did not feel like they could give themselves the opportunity to participate. Their need to manage operational complexities while at the same time leading change [25] overrode their capacity to be involved in establishing their own wellness. Leaders are expected to resiliently lead change that is intended to transform the system [25–27]. Strategies and supports for leader resilience are not elaborated in the promotion of healthcare change [28]. Instead, leaders learn, or are trained, in change processes with minimal attention paid to

the related stresses. Leadership is shaped by the expectations of healthcare policies but a leaders practices are created by the way they relate to their colleagues [29,33]. The descriptions of the leader role as a seven-day a week job suggest similarities to other studies that found a high propensity for leader burnout [e.g. 4]. The discourse around the need for staff wellbeing programs as a response to the pandemic has a focus on healthcare workers, not leadership [12–14]. The experiences of leaders in this study suggest a need for leadership to be actively included in these calls for support.

## Strengths and limitations

A strength of the study is the consistency of experiences and descriptions of leadership practice by both leaders and healthcare workers that were confirmed in co-analysis with study participants, providing confidence in the findings. However, the experiences of people who did not participate in SEED are not represented. Its not known whether they did not have an opportunity, did not think it was worthwhile or did not experience the type of leadership described by study participants. Future research should examine the perspectives of those not involved in workplace wellness programs including their experiences with leadership practices. The impact of the SEED program and related leadership practices on staff motivation and patient quality and safety has not been evaluated. Further research is required to investigate the objective results of the SEED program.

## Conclusion

Leadership practices are critical to the implementation of healthcare wellness programs. The implementation of SEED at a time of unprecedented crisis gave leaders and healthcare workers opportunities to experience something new including leadership that was courageous, responsive and authentic. Critically, the study highlighted the need for workplace wellness programs to intentionally include leaders rather than only expect them to implement them. The practices documented in this study provide guidance to others developing, implementing and researching workplace wellness programs.

## Supporting information

**S1 Checklist. COREQ (COnsolidated criteria for REporting Qualitative research) checklist.**
(PDF)

## Acknowledgments

The authors would like to acknowledge the important contributions of Chris Degeling, Summer Finlay, Sue-Ann Cutmore and Krissy Falzon to the *Narratives of Recovery—Practices supporting community mental health and well being post bush fires and COVID-19* Study that underpinned this paper.

## Author Contributions

**Conceptualization:** Julaine Allan, Katarzyna Olcon, Mim Fox, Padmini Pai, Maria Mackay, Lynne Keevers.

**Data curation:** Katarzyna Olcon, Ruth Everingham.

**Formal analysis:** Julaine Allan, Katarzyna Olcon, Ruth Everingham.

**Funding acquisition:** Julaine Allan, Katarzyna Olcon, Mim Fox, Padmini Pai, Maria Mackay, Lynne Keevers.

**Investigation:** Julaine Allan, Katarzyna Olcon, Mim Fox, Padmini Pai, Maria Mackay, Lynne Keevers.

**Methodology:** Julaine Allan, Katarzyna Olcon, Mim Fox, Lynne Keevers.

**Project administration:** Ruth Everingham.

**Validation:** Julaine Allan, Katarzyna Olcon.

**Writing – original draft:** Julaine Allan, Katarzyna Olcon, Ruth Everingham, Mim Fox, Padmini Pai, Maria Mackay, Lynne Keevers.

**Writing – review & editing:** Julaine Allan, Katarzyna Olcon, Ruth Everingham, Padmini Pai, Maria Mackay, Lynne Keevers.

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
