## [Decision Letter · Decision Letter 0]

24 Feb 2023

PONE-D-23-00842Leading wellness in healthcare: A qualitative study of leadership practices for wellness in hospital settingsPLOS ONE

Dear Dr. Allan,

Thank you for submitting your manuscript to PLOS ONE. After careful consideration, we feel that it has merit but does not fully meet PLOS ONE’s publication criteria as it currently stands. Therefore, we invite you to submit a revised version of the manuscript that addresses the points raised during the review process.

Please take on board all reviewer and editor comments which will strengthen this manuscript.

We look forward to receiving your revised manuscript.

Kind regards,

Elizabeth McGill

Academic Editor

PLOS ONE

Journal Requirements:

"This study was funded by an Australian National Health and Medical Research Council MRFF grant - 2020 COVID-19 Mental Health Research."

"JA, KO, PP, MF, MM and LK all received funding for this study from the Australian National Health and Medical Research Council Medical Research Future Fund - COVID-19 mental health research. Grant number APP2005659.

https://www.health.gov.au/summary-of-mrff-grant-recipients 

"I have read the journal's policy and the authors of this manuscript have the following competing interests: PP and RE worked for the Illawarra Shoalhaven Local Health District where this research was conducted."

Additional Editor Comments:

Please take on board all of the reviewers' comments when revising the manuscript. In addition, please address the following:

- Describe in more detail how practice theory informed the approach

- Include a table of participant characteristics

- Include more of a description of the co-analysis sessions; how were these sessions run, what was their content, etc.

- Better situate the findings from this study within the broader literature - this is briefly done in the Discussion (p.21) but should be expanded

- Include the COREQ checklist as a supplemental file

Reviewers' comments:

Reviewer's Responses to Questions

**Comments to the Author**

1. Is the manuscript technically sound, and do the data support the conclusions?

Reviewer #1: Yes

Reviewer #2: Yes

2. Has the statistical analysis been performed appropriately and rigorously? 

Reviewer #1: Yes

Reviewer #2: N/A

3. Have the authors made all data underlying the findings in their manuscript fully available?

Reviewer #1: Yes

Reviewer #2: Yes

4. Is the manuscript presented in an intelligible fashion and written in standard English?

Reviewer #1: Yes

Reviewer #2: Yes

5. Review Comments to the Author

Reviewer #1: 1) What were the experiences of the leaders in the study? For example, did most leaders in the study have over 5 or 10 years experience? The study results may differ depending on the experience of leaders.

2) Try not to use the same quote twice (see page 10).

3) Did the study look at wellness and its impact on staff motivation, patient quality and safety through leadership? It would be insightful to have this discussion.

4) Innovations were not clear in the study. Perhaps give a few examples of the innovations resulted from leadership in SEED.

5) Overall, the techniques shown by the leaders and healthcare workers were insightful in the study. The next study could look at the leadership skills in SEED impacting on staff morale, patient care, teamwork, and the organisation effectiveness.

Reviewer #2: Thank you for giving me the opportunity to read this well written paper. The topic is highly relevant also for an international audience. My main (minor) concern is that the paper, because it is a sub study of a larger project, unknown to me, it sometimes become a little intern. My advice is therefore that you:

Unfold what the SEED program consisted of, how the different versions developed (and continued)

Describe and argue in more depth for your qualitative approach (why different methods and approaches),

Describe how the groups were combined (leaders only or mixed and why)

6. PLOS authors have the option to publish the peer review history of their article (what does this mean?). If published, this will include your full peer review and any attached files.

Reviewer #1: **Yes: **Professor Huong Le-Dao

Reviewer #2: No

---

## [Author Response · Author response to Decision Letter 0]

28 Feb 2023

PONE-D-23-00842 - Leading wellness in healthcare: A qualitative study of leadership practices for wellness in hospital settings.

Thank you for the opportunity to revise and improve our manuscript. Responses are detailed in the response to reviewers document and in tracked changes in the manuscript.

---

## [Editor Report · Decision Letter 1]

17 Apr 2023

Leading wellness in healthcare: A qualitative study of leadership practices for wellness in hospital settings

PONE-D-23-00842R1

Dear Dr. Allan,

We’re pleased to inform you that your manuscript has been judged scientifically suitable for publication and will be formally accepted for publication once it meets all outstanding technical requirements.

Kind regards,

Elizabeth McGill

Academic Editor

PLOS ONE

Additional Editor Comments (optional):

Thank you for taking on board the editor and reviewer comments which have significantly improved the manuscript. I look forward to seeing this published. 
---

## [Editor Report · Acceptance letter]

19 Apr 2023

PONE-D-23-00842R1 

Leading wellness in healthcare: A qualitative study of leadership practices for wellness in hospital settings 

Dear Dr. Allan:

I'm pleased to inform you that your manuscript has been deemed suitable for publication in PLOS ONE. Congratulations! Your manuscript is now with our production department. 

Kind regards, 

on behalf of

Dr. Elizabeth McGill 

Academic Editor

PLOS ONE